# Implementation and evaluation of an electronic consult program at a large academic health system

**Anita Arora**[1]*, **Renee Fekieta**[1], **Erica Spatz**[1], **Brita Roy**[2], **Karla Marco**[1], **Mona Sharifi**[1], **Dinesh Pashankar**[1], **Babar Khokhar**[1]

**1** Yale School of Medicine, New Haven, Connecticut, United States of America, **2** NYU Grossman School of Medicine, New York, New York, United States of America

* Anita.Arora@yale.edu

## Abstract

### Background

Although the benefits of eConsults in increasing access and lowering unnecessary utilization have been well described, the development of a successful program can be challenging.

### Objective

We sought to share the experiences of a large academic health system in implementing and evaluating a high-volume electronic consultation (eConsult) program across 34 adult and pediatric medical and surgical specialties.

### Methods

Using a multi-method approach, we collected qualitative and quantitative data on operational and process outcomes to describe adoption of eConsults, and survey data to capture satisfaction and experience among referring and specialist clinicians.

### Results and conclusions

Data evaluating this eConsult program demonstrated robust uptake of the eConsult workflow as well as high satisfaction amongst primary care and specialty clinicians. Effective implementation strategies included engaging leadership, building a dedicated team, and developing quality assurance mechanisms. These experiences and findings may inform implementation at health systems interested in eConsult programs.

## Introduction

Health systems are increasingly incentivized by federal, state, and commercial payors to improve access and quality, while lowering cost of care [1]. In this setting, electronic

**Data Availability Statement:** All relevant data are within the manuscript.

**Funding:** The author(s) received no specific funding for this work.

**Competing interests:** The authors have no competing interests.

consults (eConsults), asynchronous exchanges between clinicians in the electronic medical record (EMR), have increasingly been used to obtain specialist input in a patient's care [2]. This tool allows referring clinicians, often primary care providers (PCPs), to provide more comprehensive care to patients, often avoiding the need for a face-to-face visit with a specialist [3–5]. Although the benefits of eConsults to patients, clinicians and health systems have been well described, implementation of an eConsult program can be complex [2, 4, 6].

In 2016, a small-scale pilot eConsult program between six specialties and one Federally Qualified Health Center (FQHC) in New Haven, Connecticut established feasibility and acceptability. Engagement with the Association of American Medical Colleges (AAMC) Project Coordinating Optimal Referral Experiences (CORE) provided support in expanding the program to additional clinical sites and specialties. AAMC CORE has facilitated eConsult implementation in more than 50 health systems nationally with the goals of improving communication and coordination between primary and specialty clinicians while optimizing efficiency of care [7]. Evaluation of these programs has demonstrated that eConsults often replace referrals for face-to-face visits with specialists, and are characterized by high satisfaction among engaged clinicians [5, 8, 9].

This paper describes the process of implementing and evaluating a large-scale eConsult program at an academic health system as part of the AAMC CORE collaborative, which may provide guidance for other systems designing and implementing their own programs.

## Methods

### Identification of key barriers and selection of a model

Our first step was to explore barriers to the expansion of the small-scale eConsult pilot program to additional specialties and primary care clinics. PCPs and specialists reflected that the workflow within the EMR was difficult to use, and the guidance to PCPs in advance of placing an eConsult was too lengthy. Specialists appreciated the opportunity to guide PCPs without the need for face-to-face visits, but noted that eConsults added work to their already busy clinical day. AAMC Project CORE provided insight into designing a program that was both easy to use and to scale, and advised on key steps for program implementation. These included engaging with leadership, building a dedicated team inclusive of experts in information technology and data reporting, collaboration with primary care and specialty care in content creation, and developing robust quality assurance mechanisms.

### Leadership engagement and incentive design

As a first step, support was garnered from health system leadership, including the Chief Executive Officer, Chief Strategy Officer, and Chief Clinical Transformation Officer, to engage in and develop an eConsult program. These leaders served as executive sponsors along with the Director of Population Health; and all provided oversight and guidance throughout the implementation process.

Support and engagement from leadership was critical in program success as it facilitated adoption across most medical and surgical specialties, and it supported an incentive structure that rewarded specialists for their valuable contributions. Members of the population health team created projections of eConsult volumes, and proposed an incentive payment structure awarded to specialists performing eConsults to the health system board for approval.

### Primary care and specialty clinician engagement

Once an incentive model was approved, the population health team approached specialists and primary care clinicians to teach them about the AAMC CORE model and requested their input on the design of our local eConsult model. When possible, this education was provided at regularly scheduled clinic meetings, and individuals from each primary care site were asked to serve as clinical champions. Specialist champions were also identified to serve as partners in the development of guidance in the eConsult orders, and to answer eConsults on behalf of their specialties. Primary care clinical site champions participated in a committee that met monthly to review new condition-specific guidance developed by specialists. They also led outreach to other primary care clinicians when new specialties were launched, led training sessions with their colleagues in primary care, and served as experts in the EMR workflows.

### Collaboration with Information Technology (IT)

As one of the barriers identified in the pilot was a burdensome eConsult workflow in the EMR, close collaboration was needed with colleagues in IT and with experts at the AAMC to develop a new order-based workflow. This order-based workflow allowed referring clinicians to select a condition and specialty-specific eConsult order that contained helpful decision support. The eConsult request then appeared in the EMR in-basket of the specialists, who then would have three business days to respond to the eConsult. In the early stages of implementation, biweekly meetings were held with the IT team focused on the naming, content, and format of each eConsult order to ensure the workflow was both easy for primary care and specialty clinicians to use and for the population health oversight team to monitor. IT partners also created tip sheets outlining the eConsult workflow, which were widely shared with PCPs and specialists.

### Engagement of key stakeholders, collection of early feedback, and adjustment of the model

In addition to C-suite leaders, the population health team met with key stakeholders, including leaders in specialty departments, clinical operations, and strategy. Several concerns were raised including liability, burden on specialists, and increased work for primary care clinicians. The eConsult program was modified to address these concerns. To address liability, the team added a statement developed with General Counsel to the bottom of all eConsultant responses acknowledging the limitations of guidance without the benefit of a physical exam. To address burden on specialists, the population health team advocated that the specialist incentive payments for responding to eConsults remain in place. To address concerns about increased work for primary care clinicians, the population health team clarified that the eConsult was an optional tool for PCPs.

### Billing for eConsults

In 2019, Medicare began reimbursing eConsults using CPT codes 99451 and 99452. Since then, commercial payers have followed suit, with more payers offering reimbursement for eConsults over time. As such, the population health team engaged additional colleagues in IT and Coding and Billing to adjust the eConsult clinical and EMR workflows to facilitate appropriate billing. These changes were also shared with specialists and referring clinicians, advising providers on how to discuss the possibility of cost-sharing for these consultative services with patients.

## Quality assurance and quality improvement

The analytics team built a custom report that daily and automatically extracted specific data about each eConsult into a reporting tool. Data fields included eConsult order date, specialty consulted, eConsult question, outcome (i.e., completed, converted to a face-face visit, or declined), specialist's response (for completed eConsults) and patient demographics. The tool allowed the population health team to perform ongoing quality assurance (i.e., timely and useful responses) for program improvement, create annual summary reports, and facilitate ad hoc report requests to meet the needs of providers.

To support timely responses, analysts created an automated daily email that alerted the population health team to eConsults that were pending past three business days, which was the maximum allowable time to respond to an eConsult. The population health team performed daily quality checks on these outstanding eConsults, including troubleshooting any system issues and following up with the appropriate specialty to ensure that the PCP receives a timely response.

To assure useful eConsult responses, a periodic clinical review of randomly selected eConsult questions and responses is completed by a clinician on the team. Feedback is given to PCPs and specialists as needed to ensure appropriate use of the program. PCPs were offered feedback on appropriate eConsult questions, use of the eConsult templates, and guidance on the creation of eConsult questions with enough detail for specialists to provide useful input. Specialists were offered feedback on use of the response templates, the quality of the responses themselves, and how to correctly bill or charge for the eConsult.

## Program evaluation—PCP and specialist experiences

To understand PCP's and specialists' satisfaction and experience with using eConsults for patient care, two anonymous surveys, each tailored to PCPs or specialists, were designed for completion within five minutes. (S1 and S2 Figs) These custom surveys were adapted from survey questions suggested by AAMC CORE with input from the population health team. Surveys were administered in spring and fall of 2022 via survey links emailed to PCPs (n = 530) and specialists (n = 107) participating in eConsults. Participants responded to statements about satisfaction with eConsults, ease of use, and ability to improve patient care. Participants rated level of agreement with statements using a 5-point Likert scale (strongly disagree to strongly agree).

## Results

### Process and operational outcomes

In June 2018, eConsults were launched for the first specialty, Cardiology. Over the next 18 months, an additional thirteen specialties were launched. During the COVID-19 pandemic, efficiency and interest amongst specialists increased, allowing for the launch of an additional seventeen specialties between January 2020 and October 2021. As of January 2024, there are eConsults across 19 adult and 15 pediatric specialties available to clinicians practicing in primary care, obstetrics/gynecology, and oncology practices. Among these referring sites, two are FQHCs, one is an employee health plan, and another is a free clinic.

Since program inception, 10,706 eConsults have been requested, of which 9,380 have been completed, 732 have been converted to face-to-face clinic visits and 594 have been declined by specialists. Average response time for eConsults is one business day. Sixty-one percent (n = 5,753) of completed eConsults were placed by clinicians at FQHCs.

**Table 1. Volume of eConsults by specialty.**

| Specialties | Adult eConsults completed since launch* | Pediatric eConsults completed since launch* | Total eConsults completed since launch* | Volume in 2023 |
|---|---|---|---|---|
| Allergy | 163 | 34 | 197 | 24 |
| Cardiology | 1,068 | 140 | 1,208 | 297 |
| Dermatology | 2,316 | 943 | 3,259 | 1,163 |
| Endocrinology | 757 | 366 | 1,123 | 292 |
| ENT (adult) | 57 | | 57 | 17 |
| Gastroenterology | 188 | 177 | 365 | 111 |
| Geriatrics | 13 | | 13 | 5 |
| Hematology | 757 | 171 | 928 | 343 |
| Hepatology (adult) | 171 | | 171 | 37 |
| Infectious Diseases (pediatrics) | | 42 | 42 | 12 |
| Maternal Fetal Medicine (adult) | 126 | | 126 | 53 |
| Nephrology | 182 | 41 | 223 | 74 |
| Neurology | 695 | 181 | 876 | 171 |
| Orthopedics (pediatrics) | | 9 | 9 | 1 |
| Psychiatry | 31 | | 31 | 15 |
| Pulmonary | 119 | 23 | 142 | 8 |
| Rheumatology | 331 | 40 | 371 | 120 |
| Sleep Medicine (adult) | 19 | | 19 | 9 |
| Spine (adult) | 23 | | 23 | 3 |
| Urology | 164 | 33 | 197 | 50 |
| Totals | 7,180 | 2,200 | 9,380 | 2,834 |

*Cumulative eConsults completed through 2023. Specialties have different start dates.

Among all completed eConsults, the majority required 10 minutes or less of specialist time to respond: 53% required 10 minutes or less, 34% required more than 10 minutes but less than 20 minutes, and 13% required more than 20 minutes. Volumes of eConsults by specialty are shown in Table 1. In 2022, 74% of eConsults were reimbursed by payors.

### Program evaluation—PCP and specialist experience

Survey response rate was 13% (n = 66/530) for PCPs and 72% (n = 77/107) for specialists (Table 2). Of PCP respondents, 92% (n = 61) reported satisfaction with eConsults and most found the guidance in eConsult orders helpful (86%; n = 57). Although most specialists (90%; n = 69) also reported satisfaction with eConsults, fewer agreed that eConsults improved communication with PCPs (47%; n = 36) or that access to appointments improved (32%; n = 25).

When asked what alternative first step PCPs would have taken if eConsults were not an option, the most common selection was to order a standard referral visit (83%; Table 3). Open-ended comments from specialists reflected a common theme for more transparent feedback on eConsult metrics (e.g., volumes by specialty, eConsult completion times).

### Discussion

We describe the process of implementation and subsequent evaluation of an eConsult program at one large academic health system. There were a few aspects to the program that were critical to the successful launch of more than 30 specialties and completion of more than 9,000

**Table 2. PCP and specialist satisfaction.**

| PCP Survey (n = 66, 13% response rate) | % (n) "strongly agree" or "agree" |
|---|---|
| I am highly satisfied with the eConsult program. | 92% (61) |
| eConsults are easy to use. | 94% (62) |
| eConsults improve my ability to treat specialty conditions. | 94% (62) |
| eConsults improve overall quality of care. | 94% (62) |
| The guidance in eConsult orders is helpful. | 86% (57) |
| **Specialist Survey (n = 77, 72% response rate)** | |
| I am highly satisfied with the eConsult program. | 90% (69) |
| eConsults improve overall quality of care. | 86% (66) |
| PCPs ask appropriate eConsult questions. | 82% (63) |
| I often do not have enough information to answer the question. | 8% (6) |
| | *(lower is better)* |
| Use of eConsults has improved my communication with PCPs. | 47% (36) |
| The eConsult program is improving access to in-person appointments in my department. | 32% (25) |
| eConsults are burdensome to address. | 12% (9) |
| | *(lower is better)* |

Note. PCPs (n = 3) who reported that they had not yet placed eConsults were excluded.

eConsults over a four-year period. First was engagement with senior leadership at the outset to acquire support for compensation. Second was the identification of clinical champions who were highly engaged in the program and provided constant feedback on the design and development of condition-specific eConsult orders. Finally, a partnership with creative and thoughtful experts in IT and Data Analytics helped create an EMR based clinical workflow that was easy to use, monitor and study. The importance of collaborating with partners in primary care, specialty care, IT and Data Analytics has been highlighted by the AAMC Project CORE as well as other health systems implementing eConsults [7, 9].

Early in implementation, clinicians raised concerns that eConsults would increase time spent in the EMR, which has been linked to clinician burnout [10]. However, the majority of eConsults required less than 10 minutes to draft a response. Further, we co-designed with PCPs an order-based eConsult workflow that aligned directly with that of a standard referral process. We also worked with PCPs and specialists on condition-specific templates that were easy to use, provided the PCP with basic guidance and support for asking a clear clinical question for that condition, and pulled in relevant information from the patient's record.

Fortunately, there has been consistently positive feedback from both PCPs and specialists engaged in eConsults. The majority of clinicians surveyed also perceived an improvement in

**Table 3. Alternatives to eConsults.**

| Consider the eConsults that you have placed. If eConsults were not an option, what would your alternate first step have been to address questions. Check all that apply. | % (n) of PCPs |
|---|---|
| Order a standard referral visit to the specialist | 83% (55) |
| Search medical reference/clinical guidelines | 52% (34) |
| Contact specialist via inbasket message in EMR | 47% (31) |
| Contact specialist via pager/phone | 18% (12) |
| Contact specialist via email (outside of EMR) | 3% (2) |

quality of care due to eConsults. These findings are consistent with other studies of eConsults noting perceived improvement in quality of care for patients and satisfaction among engaged clinicians [9, 11]. It was also noted that 94% of PCPs said that eConsults improved their ability to treat specialty conditions. A multi-center study of PCP experience with eConsults noted a similar finding, with PCPs reporting increased knowledge and comprehensiveness of primary care through eConsults [5]. Interestingly, only half of specialists noted that communication with PCPs was improved as a result of eConsults. Qualitative assessment may be needed to better understand the barriers to communication and how to use eConsults to strengthen communication.

In reflecting on eConsults they had placed, 83% of PCPs noted that they would have ordered a standard referral in place of at least one eConsult. This finding aligns with conclusions from other health systems that eConsults reduce unnecessary referrals to specialty care and suggests potential cost savings to both patients and payors [3].

In conclusion, engagement with a national eConsult learning collaborative led by the AAMC, highly engaged leadership, clinical champions and strong partners in IT and Data Analytics helped to create a sustainable eConsult program, with high satisfaction among both referring and specialty providers. Lessons learned at this large academic health system may serve as a guide for other sites in their design and implementation of eConsult programs.

## Supporting information

**S1 Fig. Primary care provider survey.**
(DOCX)

**S2 Fig. Specialist survey.**
(DOCX)

## Author Contributions

**Conceptualization:** Anita Arora, Renee Fekieta, Erica Spatz, Brita Roy, Karla Marco, Mona Sharifi, Dinesh Pashankar, Babar Khokhar.

**Data curation:** Renee Fekieta.

**Formal analysis:** Renee Fekieta.

**Methodology:** Anita Arora, Renee Fekieta, Erica Spatz, Brita Roy, Karla Marco, Mona Sharifi, Dinesh Pashankar, Babar Khokhar.

**Writing – original draft:** Anita Arora, Renee Fekieta, Erica Spatz, Brita Roy, Karla Marco, Mona Sharifi, Dinesh Pashankar, Babar Khokhar.

**Writing – review & editing:** Anita Arora, Renee Fekieta, Erica Spatz, Brita Roy, Karla Marco, Mona Sharifi, Dinesh Pashankar, Babar Khokhar.

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
