## [Decision Letter · Decision Letter 0]

26 Jun 2024

PONE-D-24-09871Implementation and evaluation of an electronic consult program at a large academic health systemPLOS ONE

Dear Dr. Arora,

Thank you for submitting your manuscript to PLOS ONE. After careful consideration, we feel that it has merit but does not fully meet PLOS ONE’s publication criteria as it currently stands. Therefore, we invite you to submit a revised version of the manuscript that addresses the points raised during the review process.

We look forward to receiving your revised manuscript.

Kind regards,

Maria Rosaria Gualano, MD, MPH

Academic Editor

PLOS ONE

Journal Requirements:

3. We note that your Data Availability Statement is currently as follows: All relevant data are within the manuscript.

Reviewers' comments:

Reviewer's Responses to Questions

**Comments to the Author**

1. Is the manuscript technically sound, and do the data support the conclusions?

Reviewer #1: Partly

2. Has the statistical analysis been performed appropriately and rigorously? 

Reviewer #1: No

3. Have the authors made all data underlying the findings in their manuscript fully available?

Reviewer #1: No

4. Is the manuscript presented in an intelligible fashion and written in standard English?

Reviewer #1: Yes

5. Review Comments to the Author

Reviewer #1: The topic is overall interesting, however some suggestions are necessary:

The abstract is very short and could be improved to be more structured and more insightful to the reader.

Please clarify what you mean by "specialty input"

With regards to the Methods section, most of the information does not belong to this section as it already narratively reports findings. This information can be shortened and reported in the Results section

Overall, a key point to be addressed is that the study design of choice (e.g.mixed-methods study?) should be made more clear from the beginning, otherwise the narrative reporting of this experience, however interesting, falls short of an appropriate structure and methodology.

With regards to the Surveys, how were they developed, were they validated? It is important to detail this process in the methods section. Also, the surveys should be made fully available.

6. PLOS authors have the option to publish the peer review history of their article (what does this mean?). If published, this will include your full peer review and any attached files.

Reviewer #1: **Yes: **Marcello Di Pumpo

---

## [Author Response · Author response to Decision Letter 0]

8 Aug 2024

Author Response Letter:

Title: Implementation and evaluation of an electronic consult program at a large academic health system

REVIEWER COMMENTS:

Reviewer #1:

1. Is the manuscript technically sound, and do the data support the conclusions?

Reviewer #1: Partly

Response: Please see modifications described in the following sections.

2. Has the statistical analysis been performed appropriately and rigorously? 

Reviewer #1: No

Response: Thank you for this point. We provided descriptive data (e.g., eConsult volumes, provider satisfaction) that was used to support implementation and evaluation of our electronic consultation (eConsult) program. We hope that our practical approach to describing our experience may help other health systems design, implement and evaluate their eConsult programs.

3. Have the authors made all data underlying the findings in their manuscript fully available?

Reviewer #1: No

Response: Thank you for bringing this to our attention. We added numerators and denominators to supplement percentages from survey data summarized in Table 2 (PCP and Specialist Satisfaction) and Table 3 (Alternatives to eConsults). No changes were necessary for Table 1 as the data are summarized in raw form to show our institution’s eConsult volumes for context.

4. Is the manuscript presented in an intelligible fashion and written in standard English?

Reviewer #1: Yes

Response: Thank you very much.

5. Review Comments to the Author

• Reviewer #1: The topic is overall interesting; however, some suggestions are necessary:

The abstract is very short and could be improved to be more structured and more insightful to the reader.

Response: We have added structure and additional detail to the abstract in lines 21-35.

“Background: Although the benefits of eConsults in increasing access and lowering unnecessary utilization have been well described, the development of a successful program can be challenging. 

Objective: We sought to share the experiences of a large academic health system in implementing and evaluating a high-volume electronic consultation (eConsult) program across 34 adult and pediatric medical and surgical specialties. 

Methods: Using a multi-method approach, we collected qualitative and quantitative data on operational and process outcomes to describe adoption of eConsults, and survey data to capture satisfaction and experience among referring and specialist clinicians. 

Results and Conclusions: Data evaluating this eConsult program demonstrated robust uptake of the eConsult workflow as well as high satisfaction amongst primary care and specialty clinicians. Effective implementation strategies included engaging leadership, building a dedicated team, and developing quality assurance mechanisms. These experiences and findings may inform implementation at health systems interested in eConsult programs.”

• Please clarify what you mean by "specialty input"

Response: Thank you for the clarification. This is referring to guidance from a specialist in patient care. We have clarified the language in lines 38-40.

“In this setting, electronic consults (eConsults), asynchronous exchanges between clinicians in the electronic medical record (EMR), have increasingly been used to obtain specialist input in a patient’s care.”

• With regards to the Methods section, most of the information does not belong to this section as it already narratively reports findings. This information can be shortened and reported in the Results section.

Response: In the Methods section under the subheading “Engagement of key stakeholders, collection of early feedback, and adjustment of the model”, we removed this section (after line 141) as we decided these early implementation data did not add value to the manuscript.

• Overall, a key point to be addressed is that the study design of choice (e.g.mixed-methods study?) should be made more clear from the beginning, otherwise the narrative reporting of this experience, however interesting, falls short of an appropriate structure and methodology.

Response: We have clarified this in the abstract: “Using a multi-method approach, we collected qualitative and quantitative data on operational and process outcomes to describe adoption of eConsults, and survey data to capture satisfaction and experience among referring and specialist clinicians.”

• With regards to the Surveys, how were they developed, were they validated? It is important to detail this process in the methods section. Also, the surveys should be made fully available.

Response: The surveys were not validated but rather developed to gather rapid feedback from providers and inform program improvement. We did a careful review of other similar surveys available from the AAMC and other sites using eConsults. As a team, we adapted these other surveys to suit our institution’s needs. In the Methods, under the subheading “Program evaluation – PCP and specialist experiences”, we added a sentence (line 254) to reflect our survey design process. We have also made the surveys available.

6. PLOS authors have the option to publish the peer review history of their article (what does this mean?). If published, this will include your full peer review and any attached files.

Do you want your identity to be public for this peer review? For information about this choice, including consent withdrawal, please see our Privacy Policy.

Reviewer #1: Yes: Marcello Di Pumpo

Response: Thank you for your careful review. We are happy to include the peer review history.

---

## [Editor Report · Decision Letter 1]

26 Aug 2024

Implementation and evaluation of an electronic consult program at a large academic health system

PONE-D-24-09871R1

Dear Dr. Anita Shinali Arora,

We’re pleased to inform you that your manuscript has been judged scientifically suitable for publication and will be formally accepted for publication once it meets all outstanding technical requirements.

Kind regards,

Maria Rosaria Gualano, MD, MPH

Academic Editor

PLOS ONE
---

## [Editor Report · Acceptance letter]

3 Sep 2024

PONE-D-24-09871R1 

PLOS ONE

Dear Dr. Arora, 

I'm pleased to inform you that your manuscript has been deemed suitable for publication in PLOS ONE. Congratulations! Your manuscript is now being handed over to our production team.

Kind regards, 

on behalf of

Dr. Maria Rosaria Gualano 

Academic Editor

PLOS ONE